# Singing Transcription from Polyphonic Music Using Melody Contour Filtering

**Zhuang He and Yin Feng \***

Department of Artificial Intelligence, Xiamen University, Xiamen 361000, China; hezhuang@stu.xmu.edu.cn

\* Correspondence: FengYin@xmu.edu.cn

**Abstract:** Automatic singing transcription and analysis from polyphonic music records are essential in a number of indexing techniques for computational auditory scenes. To obtain a note-level sequence in this work, we divide the singing transcription task into two subtasks: melody extraction and note transcription. We construct a salience function in terms of harmonic and rhythmic similarity and a measurement of spectral balance. Central to our proposed method is the measurement of melody contours, which are calculated using edge searching based on their continuity properties. We calculate the mean contour salience by separating melody analysis from the adjacent breakpoint connective strength matrix, and we select the final melody contour to determine MIDI notes. This unique method, combining audio signals with image edge analysis, provides a more interpretable analysis platform for continuous singing signals. Experimental analysis using Music Information Retrieval Evaluation Exchange (MIREX) datasets shows that our technique achieves promising results both for audio melody extraction and polyphonic singing transcription.

**Keywords:** singing transcription; audio signal analysis; melody contour; audio melody extraction; music information retrieval

## 1. Introduction

The task of singing transcription from polyphonic music has always been worthy of research, and it represents one of the most difficult challenges when analyzing audio information and retrieving music content. The automatic music transcription approach described in [1] exclusively considers instrumental music and does not discuss the effects of drum instruments. The emphasis is on estimating the multiple fundamental frequencies of several concurrent sounds, such as a piano. The definition of polyphonic music is different from multiple fundamental frequency estimation and tracking [2] and automatic music transcription. In music, polyphony is the simultaneous combination of two or more tones or melodic lines. However, according to the description in Mirex2020 Singing Transcription, the task of transcribing polyphonic music containing only monophonic vocals into notes is known as singing transcription from polyphonic music [3]. Even a single interval made up of two simultaneous tones, or a chord of three simultaneous tones, is rudimentarily polyphonic. A piece of music could exhibit a rather moderate degree of polyphony, featuring a predominant singing voice and a light accompaniment. This definition is consistent with the description in [4], and the singing voice is the source of the main melody in popular music. The data collections used for the evaluation of melody extraction tasks and singing transcription tasks deal mostly with percussive instrument and rhythmic instrumental music.

In this article we divide singing transcription into two sub-tasks based on note-level and frame-level factors. First, from the perspective of notation, note transcription, with notes as the basic units, is consistent with people's cognition of music. Second, according to human auditory effects, melody extraction, which uses signal frames as the basic units, conforms to the characteristic of decomposing audio into signal frames through a Fourier transform in audio analysis. The audio signal displayed on the two-dimensional plane is

consistent with human subjective cognition on the time axis, with the loudness of multiple signals superimposed on the vertical axis. The direct processing of notes-level signals seems to be inapplicable to singing transcription systems. Extracting a signal frame that represents the main melody and converting it to a symbol representation similar to MIDI makes it easier to understand [5]. The signal frame acts as an intermediary to connect audio frequencies and notes. Therefore, the melody contour extracted in our paper can be used to directly obtain the signal frame, and the basic information composed of note-level factors can also be obtained by segmenting and calculating the data. Since singing transcription has become an official project of MIREX this year, it has been given a clear definition of singing transcription and the test collections of "Cmedia". There are a few algorithms available for the evaluation and verification of singing transcription tasks. Therefore, we use the performance of melody extraction at the frame level to further evaluate the accuracy of our algorithms. Extracting a melody from polyphonic music is the process of extracting the fundamental frequency from the singing signal. Finding fundamental frequencies from complex audio signals with an accompaniment, in addition to the analysis of the contour characteristics, can help us achieve the appropriate reproduction mode for music research. The extracted melody can be used for music information retrieval such as humming recognition [6], a karaoke environment for a melody line to verify the accuracy of a singer [7], or to maintain originality and effectively prevent copyright issues [8,9]. It even be used in secondary creation to achieve more interesting results, such as background music (BGM).

A number of methods have been presented regarding the analysis of signal frames. Over the past decade, a series of excellent algorithm applications and proposals have brought about great improvements to melody extraction tasks, and more results have been referred to MIREX from 2005 to present. The main points is to extract the melody of the singing voice by signal processing, according to the spectrum diagram [10–13], which is mixed with the separation algorithm [14] and non-negative matrix factorization (NMF) [15]. Due to the widespread use of neural networks [16], the results have reached a high level of maturity.

Our observation and research found that an accompaniment affects fluctuation performance as significantly as a singing voice, specifically in a karaoke scene. In this situation, a previously present singing voice melody was extracted from a complex mixture in which the melody was embedded between two masking melodies—one with a lower accompaniment and the other with a higher fundamental frequency. Two main challenges were presented by this task: first, the estimation of the fundamental frequency corresponding to the singing voice melody [3,17], and second, the elimination of the interference of octave errors [18]. In [17,19], the existence of a singing voice was judged by a salience function while, in most cases [16,20], this has been detected with a neural network. The two key components of our method are the extraction of edge features from an input audio mixture and melody contour filtering from overall contour lines. The system shows significantly improved performance compared to previous approaches.

While previous studies on singing transcription have emphasized monophonic music genres [21,22] or a single accompaniment [23], the present method focuses on a more robust performance, with an elaborate configuration according to the auditory environment. The proposed algorithm originates from an audio signal and relies on the use of information from the spectrum graph to detect all of the melodies. Therefore, it can be seen that all melody extraction algorithms are inseparable from the feedback results of information in the frequency domain generated by the Fourier transform. We observe the distribution of the singing voice through the spectrum diagram and try to find a certain distribution law from it to facilitate the implementation of our algorithm [24]. We demonstrate that a great deal of information from the signal can be discarded while still retaining its rhythmical aspects. The creation of a singing transcription system that is able to transcribe polyphonic music in a karaoke scene, without setting restrictions on the degree of polyphony and instrument types, is still an open question.

The rest of the article is structured as follows. In Section 2, the phase spectral reconstruction using sinusoid extraction and the building of the salient spectrogram by harmonic superposition are presented. The melody contours are selected by a salience function based on edge searching and contour filtering by setting breakpoints and post-processing. In Section 3, the melody contour is segmented using stability region segmentation and a pitch line matching algorithm. In Section 4, we evaluate our methods by MIREX audio melody extraction metrics and the singing transcription criterion. Finally, in Section 5, we present conclusions about our system.

## 2. Melody Extraction

The block diagram in Figure 1 illustrates an overview of the proposed algorithm. The following subsections describe successive stages of the singing transcription system.

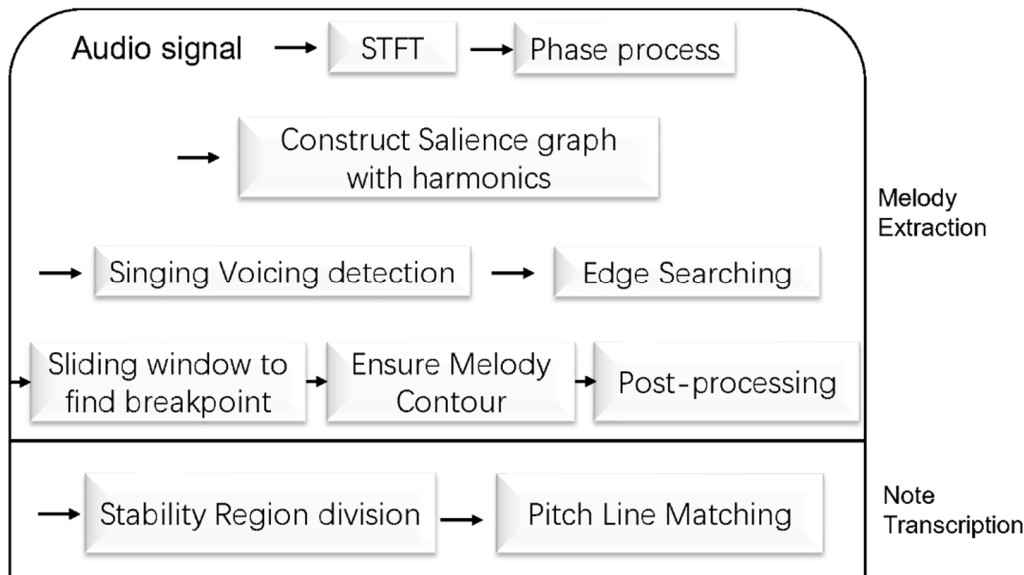

**Figure 1.** Block diagram: melody extraction and note transcription methods constituting the singing transcription system.

### 2.1. Phase Processing

In the audio signal processing field, Fourier transform technology, which is used for converting the time-frequency domain, has been widely used. With the development of preprocessing by many open-source projects, including those by Librosa [25] and Madmom [26], additional preprocessing of input audio signals will no longer be necessary, such as enhancing high-frequency-band energy with a high-pass filter [10]. The advantage of this is that it is only necessary to adjust the parameters and change the function to obtain the optimal result, which improves the program's running efficiency and reduces the difficulty of basic analysis. One of the most classical and enduring formulas is the discrete Fourier transform, which is formulated as follows,

$$X(l, k) = \sum_{k=0}^{N-1} w(n) x(n + l * H) e^{\frac{-j2\pi kn}{N}} \tag{1}$$

where $k \in \left(-\frac{N}{2}, \frac{N}{2}\right), l = 0, 1, 2\ldots$, in the sequence; $x(n)$ is the input audio signal; w(n) is the window function; N is the number of points for the FFT (fast Fourier transform); H is the hop length; and $l$ is the number of frames. The original signal sampling rate is 44.1 KHz, $n = 8192$, H = 441 (satisfying the requirement of 10 ms for a frame; the overlap rate is 94.6%). For a larger window length and a higher frequency resolution, the relative time resolution is comparatively reduced. Effective specification information that involves

a relatively small hop length and large window length facilitates the subsequent detection effect of spectral edge searching.

The accuracy of the FFT bin frequency depends on the sampling rate and the number of points in the FFT, N. According to the Nyquist sampling theorem, each bin represents 5.38 Hz. The most relevant information contained in the spectrogram resulting in sinusoids noise is included in the spectral data analysis. Due to the logarithmic relationship of frequency $f_0$ and pitch,

$$p = 12log_2\left(\frac{f_0}{440}\right) + 69 \tag{2}$$

We need to improve the accuracy of the low frequency more effectively; thus, it is necessary to calculate the instantaneous frequency and amplitude using the Fourier coefficient. The instantaneous frequency calculation proposed by Boashash [27] has been used as a classical algorithm, and the FFT phase spectrum based on the estimation method in [28] can also provide a good reference for the analysis of polyphonic music.

We use a famous method, phase vocoder, to calculate the instantaneous frequency and amplitude. However, the calculation of amplitude is slightly different from the traditional approach: we processed the phase angle of the neighboring frame in the spectrogram graph obtained by a Fourier transform with the phase Angle $\varphi_l$ of the previous frame $\varphi_{l-1}$, which is calculated as follows:

$$f_l = \Gamma'\left(\varphi_l - \varphi_{l-1} - \frac{\pi H}{N}k\right) \tag{3}$$

$$\Gamma' = \frac{2l_k - l_{max} - l_{min}}{l_{max}} \tag{4}$$

where $k = 0, 1 \ldots \frac{N}{2} - 1$; $\Gamma'$ is the constructed normalization function; and the phase angle difference is within a stable distribution within the interval $(-2\pi, 2\pi)$. With a gradual increase in the $k$ value, the high-frequency part affected by phase angle change is reduced, and the opposite phase angle change is larger in the low-frequency area. $\Gamma'$ can clearly reflect the rate of phase angle change, and the function effectively results in fluctuations in radiation in the phase angle difference, reflecting the low-frequency area between adjacent frames of instantaneous frequency change. The instantaneous frequency and amplitude are calculated as follows:

$$\widetilde{A}_l = \frac{A_l sinc(\delta \pi f_l)}{1 - (\delta f_l)^2} \tag{5}$$

$$\widetilde{F}_l = \sigma f_l + \frac{f_s}{N}k \tag{6}$$

where $\delta = 0.24$, $\sigma = 1.5$, $A_l$ represents the origin amplitude of each bin, and the amplitude of each bin is recalculated through the designed kernel function to obtain the instantaneous amplitude $\widetilde{A}_l$. This function reduces the energy of both low-frequency and high-frequency regions in proportion to the reduction in the error of subsequent melody extractions caused by excessive energy in the accompaniment and octaves. The extraction of the instantaneous frequency $\widetilde{F}_l$ is the superposition of the frequency $f_l$ and each bin. In the next step, we only retain the instantaneous frequency and amplitude in the spectrum, and the non-peak points are filtered in the new spectrum. In this way, we preliminarily retain the main features of the spectrogram through the Fourier transform and reconstruct the information for the peak points.

### 2.2. Construct Salience Spectrogram

In this section, we reconstruct the entire energy frequency distribution by constructing a salience graph. Similar to the salience function [29] in some classical algorithms, the energy of the higher harmonics is accumulated downward in an attempt to enhance the energy of the fundamental frequency.

Next, we convert the instantaneous frequency into a MIDI pitch. In order to better match the auditory range of the human voice, we set the pitch range to five octaves from

A1 to A6, where the singing voice is at an effective level, which is divided into 60 semitones. Analogous to the salience function used by Salamon [17], we covered the frequency range from 55 to 1760 Hz, implying that our bin range was from 11 to 352, while the quantization range was reduced by a factor of 10. According to the human auditory effect, to judge the standard of a pitch change when the fluctuation of adjacent frames is more than a specific threshold, a new note appears. Similarly, the difference in pitch between adjacent peaks is allowed to be within one semitone, which is analogous to the distance between adjacent pixels on the edge of an image. It is this pitch difference that is an important indicator for distinguishing between melodic contours. Therefore, in analyzing melodic properties, we classify pitches as semitones. Our formula for obtaining the relevant MIDI pitch for each peak point of each frame of $\widetilde{l}_k$ is as follows:

$$B\left(\widetilde{l}_k\right) = 12 log\left(\frac{\widetilde{l}_k}{55}\right) + 0.1 \tag{7}$$

After the frequency is converted to a MIDI pitch, we use the salience function to conduct a down-scaling superposition of the octave at the peak point $\widetilde{l}_k$. The energy of the high-order harmonic superposition gradually decreases with the increase in harmonic frequency. The equation is as follows:

$$F(\tau) = \sum_{k=1}^{I} \sum_{h=1}^{N} \begin{cases} \widetilde{l}_k cos^2\left(\frac{\pi}{2}\delta\right)\alpha^{h-1}, \ if \ |\delta| \leq 1 \\ \qquad\qquad 0, \ else \end{cases} \tag{8}$$

where $\delta = \left|B\left(\frac{\widetilde{l}_k}{h}\right) - \tau\right|$; $I$ corresponds to 60 bins; N represents the count of harmonics; the experimental result is 10; $\tau = 1, 2, 3\ldots$; and $\alpha$ is the parameter of the high-order harmonic attenuation coefficient, which is chosen as 0.8. The judgment condition represents $\widetilde{l}_k$, the condition that the absolute value of the pitch difference between each octave of the peak point and the fundamental frequency is less than a semitone. The obtained peak energy is taken as the cosine change multiplied by the power of the number of harmonics. The cosine function means that the result of the corresponding salient function is a fluctuation distribution, which is conducive to the superposition between peak values, and it increases the convenience of our later contour search problems. The result also shows a remarkable effect in terms of finding the potential fundamental frequency and dealing with the misjudgment caused by octave error. The superimposing of harmonic energy can also effectively increase the energy at some peak points, stabilize and enhance the energy distribution of the system, and contribute to the accuracy of detecting a singing voice.

Through further analysis of a large number of graphs, the energy contained in the low-frequency region has a magnitude range substantially higher than most of the medium-frequency regions. In order to maintain the relative balance of bands in different frequency domains, we use a Gaussian weighting function for each frame to reduce the energy in the low-frequency region.

$$G(\tau) = F(\tau)e^{-\delta\left(\frac{k}{I} - 0.5\right)^2} \tag{9}$$

where $k = 0, 1, 2\ldots$ and $\delta = 3.3$; weighting the low-frequency harmonics separately is helpful for enhancing the fundamental frequency result, allowing us to effectively control the overall realization of low-frequency suppression. Excessive low-frequency energy is the main factor causing the potential deviation of the fundamental frequency [30]. In the case of similar energy regions, weighting can make the selection of the fundamental frequency more centralized, which is in line with the human auditory effect. We chose Gaussian weighting due to its versatility and high level of implementation, and the robustness of the system was therefore enhanced.

### 2.3. Singing Voice Detection

After the construction and processing of the salience graph, we filtered out spectral peak in each frame to find the singing voice. We applied edge searching [31], which is widely used in image recognition to find potential contours. Two types of characteristic—length and energy—can be used to determine a melody.

In the reconstructed salience graph, the frequency and peak energy correspond to a MIDI pitch and instantaneous amplitude, respectively. Each frame contains multiple peaks as valid information. Two problems are still present in melody extraction: voice detection and octave error [17]. At this time, the frequency of the accompaniment, the number of singing voices, and many other interference sources exist in the overall graph. In contrast to automatic singing transcription from a monophonic signal [32], the presence of an accompaniment presents a huge challenge. The salience spectrum constructed in the previous section highlights the features and quantifies the parameters more accurately; thus, a solution to this dilemma is presented.

On the basis of the salience graph, we first we calculate the energy mean $\mu$ and the standard deviation $\sigma$ of the peak points in the whole graph.

$$\sigma = \sqrt{\frac{\sum_{i=1}^{n}(S_i - \mu)^2}{n}} \tag{10}$$

We calculate the threshold value of the human voice as $v = \mu - \theta\sigma$, where $\theta = 0.5$. The maximum energy of peak points in frames $S^+$ less than $v$ will be recorded as non-voice frames $S^-$. Frame-level energy filtering seems to be the simplest and most efficient implementation in singing voice filtering.

Similarly, we filter the peak point $S < \mu S^+$ in each frame and take the smaller parameter $\mu = 0.3$. To some extent, this step is beneficial for removing the smaller values and reducing excessively long contour disturbances in edge searching. The interference caused to the new contour by the last sound of the previous one is blocked by the discontinuity point in the frame. This kind of interference is a problem, as the long continuation of the previous contour may overlay the new contour. Using the features of the edge searching algorithm, it is not possible to determine a highly continuous contour profile from global analysis. Thus, the study of the breakpoint is a critical step for our entire experiment. In the following sections, we also propose new algorithms to solve the problem of excessively long contours.

### 2.4. Edge Searching

As the basic feature of an image, the contour is the essential element of object detection and segmentation. Similarly, for the salience graph, we search each contour, filter most invalid information, and improve the SNR (signal-to-noise ratio) of the overall graph, which are all important steps. Coincidentally, the energy of the singing voice is distributed beyond that of the accompaniment, and it is distinct and clear. The method of edge searching for the entire graph proceeds as follows:

- Binarization is performed for all peak points and classified as $S^+$;
- The distance between two peak points in neighboring frames must not be greater than $\sqrt{2}$ of that of the continuous contours;
- When two points $\left(S_i^+, S_{i+1}^+\right)$ are discontinuous, the corresponding contour point $S_{i+1}^+$ is generated as a new contour points, and $S_i^+$ is the end of the previous contour;
- Until the completion of the entire salience graph search, the process is repeated.

Based on this searching principle, we extract all the edges $\Omega_E$ in the graph. The peak points in the middle region are used as the characteristics of their contours. For each $\Omega_E$, there are two important properties: length and average energy. Length is used to ensure that the length of the shortest note is not less than a fixed threshold $\alpha$.

For either $E_l \in \Omega_E$, $Len(E_l) < \alpha$ is filtered, the constructed spectrum presents a sparse distribution, and the SNR is consequently higher. As shown in Figure 2, the contour information in the figure is still rich.

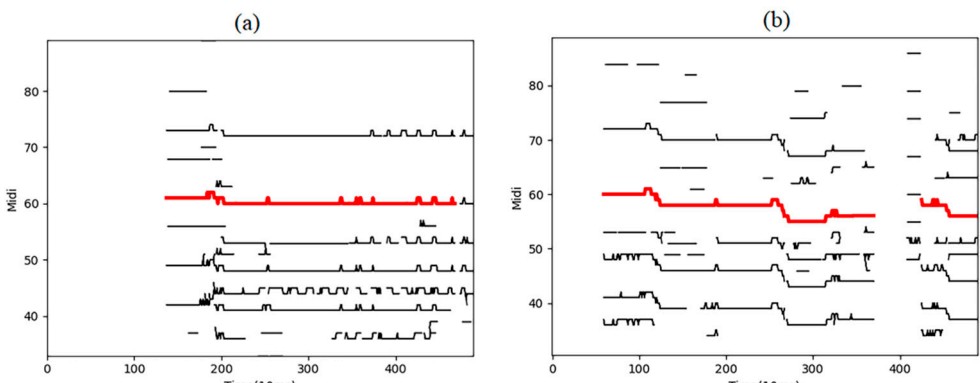

**Figure 2.** Red lines represent the contours of the human voice, while black lines represent the contours of interference: (**a**) ADC04.daisy01.wav, (**b**) ADC04.daisy02.wav.

Next, we look for the breakpoints in the contours of different syllables to cut them so that the contours are as close to the length of a note as possible without falling below the threshold $\alpha$.

### 2.5. Melody Contour Filtering
#### 2.5.1. Sliding Window

In Section 2.4, we proposed a method for intra-frame breakpoint screening to remove interference caused by long endnotes or homophonic accompaniment. Unfortunately, the aim of intermittent contours was not achieved between adjacent frames. In this section, we use the sliding window algorithm to find the breakpoints in all the contours $\Omega_E$. In essence, the subtle fluctuation in energy generated by two adjacent notes at the moment of a sound change is the change in syllables that produces one note after another. Our algorithm determines the energy difference in the change in syllables. The sliding window method is as follows:

(1) Superpose the peak points $\widetilde{l}_k$ in each frame to obtain the total energy of a frame $E_l$ and create a difference in the energy difference axis, as shown in Figure 3;
(2) Select a window $W$ with a specific length, and the hop length is $W/2$ padded with zero, while the length of the tail window is insufficient;
(3) Search $E_{max}$ in each window and constitute a collection $\Omega_E$;
(4) When the same $E_{max}$ appears in two neighboring windows as the local maximum, all of the $E_{max}$ set a new collection $\widetilde{\Omega}_E$.
(5) Repeat the above steps (3)–(4) until all searches for the salience graph are completed.

The same $E_{max}$, which occurs in two successive windows, is considered to be a breakpoint. However, the actual minimum length is 1.5 $W$. The purpose of the algorithm is to find the local minimum value in a certain region considered to have no breakpoint; if there are multiple local minimum values, we search the next region. At this time, the corresponding frame is regarded as the breakpoint $E_{min}$ of all the contours containing the frame to segment the melody. After our experiments, the best 50 $W$ values were selected.

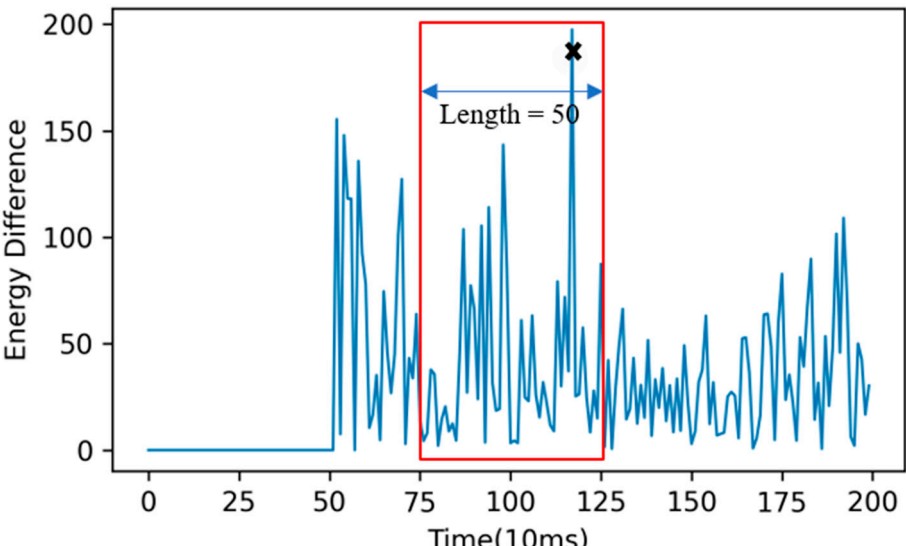

**Figure 3.** Sliding window filtering used for searching for the maximum energy difference to obtain breakpoints. The red box is our sliding window of 50 min length.

### 2.5.2. Contour Filtering

Energy characteristics in contours are regarded as the main feature in melody searching. In this section, the interfering contour are largely eliminated first; afterwards, the unique melody contour are determined within each minimum frame. Under the segmentation of $\widetilde{\Omega}_E$ (which can be regarded as filtering all the peak points in the corresponding frame and temporarily identifying them as non-voice frames), we conduct edge searching on the graph again.

We calculate the average energy $\overline{E}_k$ of the peak point of the salience graph and calculate the average energy $\overline{E}_l = E_l / len(l)$ of each contour. The contour energy $\overline{E}_l < \overline{E}_k$ is filtered, and the remaining contours are shown in Figure 4. Next, we need to select the final contours with the most obvious features in each region from $\widetilde{\Omega}_E$.

Obviously, there are multiple contours between adjacent breakpoints (shown in Figure 4), including accompaniment contours, singing voice contours, and octave contours. We selected the range between two neighboring breakpoints in $\widetilde{\Omega}_E$ to facilitate the comparison of the same melody contours and ensure the fundamental frequency of the human voice. In the previous construction of a salience function, we included the downward superposition of high-order harmonics, which increases the energy of the fundamental frequency with an abundant octave frequency to the maximum extent. Our proposed contour filtering algorithm is as follows:

(1) Compare the size of each contour $E_l$ in the region $\left( \widetilde{\Omega}_k, \widetilde{\Omega}_{k+1} \right)$ to find the contour $\widetilde{E}_l$ with the maximum energy (shown in Figure 5, the red lines);

(2) Set the head and the tail from $\widetilde{E}_l$ as $i, j$ to filter the remaining peak points in the region $\left( \widetilde{\Omega}_i, \widetilde{\Omega}_j \right)$;

(3) If either the head $i$ or tail $j$ end of their distance between the boundaries $\left( \widetilde{\Omega}_k, \widetilde{\Omega}_{k+1} \right)$ is greater than the note threshold $\alpha$, contour filtering will be conducted on the region $\left( \widetilde{\Omega}_k, \widetilde{\Omega}_i \right) \vee \left( \widetilde{\Omega}_j, \widetilde{\Omega}_{k+1} \right)$ to find the next largest contour $\widetilde{E}_{lp}$.

(4) Otherwise, the melody contour searching ends in this area and the method proceeds to the next region $\left( \widetilde{\Omega}_{k+1}, \widetilde{\Omega}_{k+2} \right)$.

(5) Repeat the above steps until all searches for the entire map are completed.

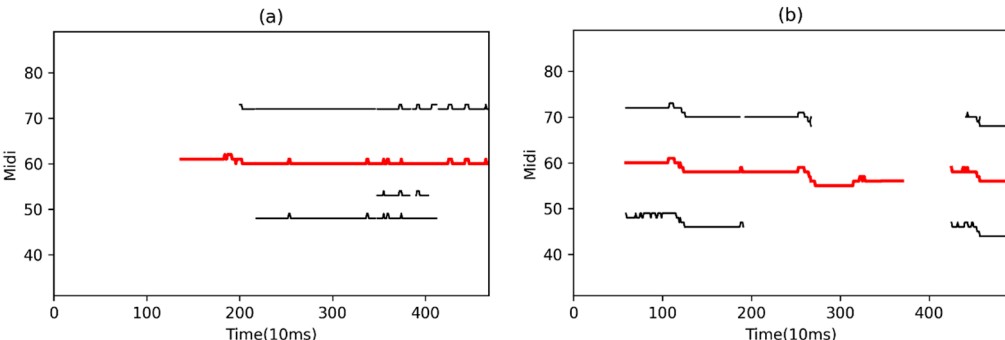

**Figure 4.** Salience graphs after average energy contour filtering. Red lines are the melody contours: (**a**) ADC04.daisy01.wav, (**b**) ADC04.daisy02.wav.

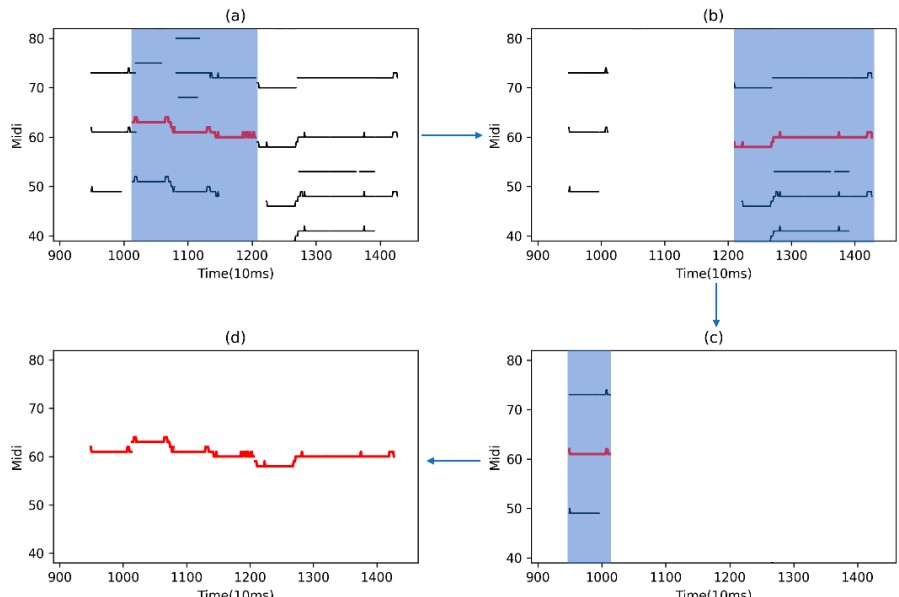

**Figure 5.** The process of melody contour filtering: (**a**) Searching the maximum energy contours on $\left(\widetilde{\Omega}_k,\ \widetilde{\Omega}_{k+1}\right)$; (**b**) searching contours on $\left(\widetilde{\Omega}_k,\ \widetilde{\Omega}_i\right) \vee \left(\widetilde{\Omega}_j,\ \widetilde{\Omega}_{k+1}\right)$; (**c**) searching the remain region; (**d**) obtaining final melody.

In the above algorithm, the regions are calculated in sections, which are filtered according to the strength of the contour energy, and multiple non-overlapping contour searches are performed on the remaining regions to determine the unique singing melody. The preprocessing of the contour filters most of the interference information, and multiple edge searches do not degrade the performance of the program.

### 2.6. Post-Processing

In the previous section, we determined the contours of each region and ensured the uniqueness of the contours through non-overlapping filtering. However, the search for the region can only determine the local optimal solution; regarding the whole, a jump in the abnormal contour cannot be avoided. Based on the human auditory effect, the notes are distributed in a fluctuating way, so we maintain relative stability at the same level. For a few anomalous contours, we introduce the concept of an abnormal contour, assuming that this leads to an anomaly and deviates from the correct path (shown in Figure 6). On this basis, we propose a post-processing method to correct abnormal contours:

(1)    Calculate the average pitch $\overline{P}$ of all current contours;

(2)    For the mean pitch $\overline{P}_l$ of each contour, see Equation (11);

(3)  If the number of abnormal contours exceeds 1/4 of the total number, do not perform contour processing.

(4)  Repeat the above steps of (2)–(3) until all contours are completed.

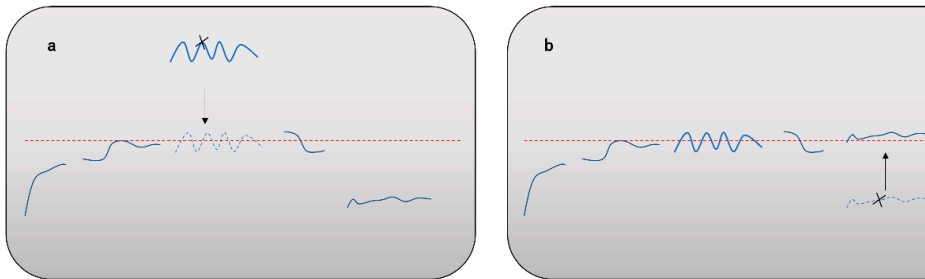

**Figure 6.** Correcting abnormal contours above or below the melody contours. The red dot line is the average pitch and blue lines are melody contours. Step (**a**) show an abnormal contour above the red dot line and step (**b**) shows an abnormal contour below the red dot line. Both of them have been corrected.

$$P_l = \begin{cases} P_l - 19, & if\ \overline{P_l} - \overline{P} > \alpha 1 \\ P_l - 12, & elif\ \overline{P_l} - \overline{P} > \alpha 2 \\ P_l + 12, & elif\ \overline{P_l} - \overline{P} < \alpha 3 \\ P_l, & else \end{cases} \tag{11}$$

According to the conversion characteristics of pitch, the double octaves and pitches are $\pm 12$ and the triple octaves and pitches are $\pm 19$. At the same time, this corresponds to the previous classification of the pitch as 60 semitones. Only the frequency doubling and the frequency tripling misjudgment interference are considered here, where $(\alpha 1, \alpha 2, \alpha 3)$ are (17,10,13), respectively.

Correcting an abnormal contour is an important step that avoids the contour jump and greatly enhances the robustness and accuracy of the program.

For polyphonic music with strong background sound, a Fourier transform will cause extremely high octave interference. The proposed post-processing method has a great effect on continuous abnormal contours, and it improved the overall result for about 5% of them.

Therefore, due to the existence of edge branches, the edge searching algorithm can only classify branches in the same contour, and the energy mean calculation includes the peak points in each branch. A fundamental pitch $p_0$ is generated by applying the strongest energy principle from each frame if additional branches exist, and then reprocessing $p_0$ is undertaken by the application of median filtering. The use of median filtering has two main advantages: the first is the reduction in the problem of pitch fluctuation caused by the selection of peak points in the branches, and the second is the filling of the breakpoints caused by the sliding window algorithm.

### 3. Note Transcription

*3.1. Two Fundamental Frequency Sequences*

Transcription on the basis of melody extraction is a process of effectively segmenting the fundamental frequency to extract notes. Unlike the multi-fundamental frequency estimation task, the audio involved is mostly generated by instrumental playing [33]. However, for the characteristics of aspiration and vibrato in the singing voice, the composition is more difficult, the boundary is less obvious, and the judgment of notes is more complicated.

After extracting the overall melody, we propose two fundamental frequency selection methods. The methods we used in both MIREX tasks are described in detail below:

- Baseline: Retain the extracted current integer frequency sequences, which only limits the input for subsequent singing transcription tasks.

- Extension: Readjust the precision of pitch to the accuracy of 10 cents (where 100 cents correspond to a semitone). The basic frequency accuracy error required for the MIREX audio melody extraction competition is within 25 cents. Therefore, we recalculate the salience function using the same instantaneous frequency and amplitude after the phase processing step. The novel pitch frequency conversion formula is as follows:

$$B\left(\widetilde{l}_k\right) = 120log\left(\frac{\widetilde{l}_k}{55}\right) + 1 \tag{12}$$

The parameters of the original function remain unchanged, producing a more accurate salience graph. After readjusting, we obtain 600 bins, and the quantization range expands 10-fold. Finally, the extracted melody results are mapped into the new graph through the following equation:

$$\widetilde{p}_0 = max\ B(10p_0 \pm \theta) \tag{13}$$

In the new salience graph, the pitch with the highest energy within the range of $2\theta$ corresponding to the semitone $p_0$ is taken as the fundamental pitch $\widetilde{p}_0$. The parameter $\theta$ is 5. The result of such a conventional mapping concept is high accuracy and a small error range. The disadvantage is that the salient function is recalculated with a span of 10 cents, which greatly increases the running time of the whole program. The extension results were uploaded to the melody extraction task of MIREX 2020. The comparative evaluation is shown in Section 4.1.

### 3.2. Stability Region Division

The composition of notes includes three features: onset, pitch, and offset. The more critical of these criteria are onset and pitch. Similar to the solo singing task, discrimination can be obtained by using the two characteristics of pitch change and energy fluctuation, and even higher accuracy can be achieved by the latter [34]. To distinguish the boundaries of two notes, we turn to the search for regions of stability. The definition of a region of stability is that each pitch fluctuation is within a certain range. Analogous to Section 2.5.1, we effectively establish window sliding conceptions for onset detection. As seen in Figure 7, our proposed stability region division algorithm is detailed as follows:

(1) The initial position $l_0$ of each contour $l$ is regarded as an onset. If $l < \theta$, the contour is indecomposable, and $l_{end}$ is regarded as an offset and is transferred to the segmentation of the next contour;

(2) Create a sliding window $W$ (as shown in the block in Figure 7), where the initial position is one window length, $P_{max}$ is the maximum pitch in the window, $P_0$ is the first pitch, and $P_7$ is the last pitch;

(3) For each window, there are following operating conditions:

    (a) If $P_{max} - P_0 > \alpha$, where the subscript of $P_0$ is denoted as $Off_0$, and the hop length is one window length;

    (b) If $P_{max} - P_7 > \alpha$, where the subscript of $P_7$ is denoted as $Off_0$, and the hop length is two window lengths;

    (c) If neither of above is true, we consider the window to have attained stability, the hop length is a frame. We then recalculate Step (3) until the distance between $P_0$ and $l_{end}$ is less than a unit length 12;

(4) Repeat steps (3) until all contour division are completed.

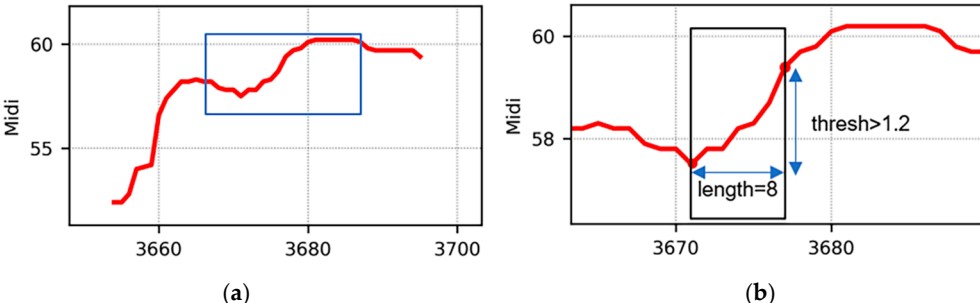

**Figure 7.** Segmentation of notes using stability region: (**a**) is a fragment of the first song in the dataset "Cmedia" from singing transcription task; (**b**) shows the detailed data in the blue box, where the black box represents the sliding window and the threshold in the figure is the optimal pitch difference of the melody segmentation.

Regarding the minimum length of a note, we consider at least eight frames to be a stable note. Simultaneously, a contour length greater than 20 frames is considered separable. We ensure that the tail note meets the minimum length condition even after segmentation. The position relationship of pitch differences is calculated to determine the syncopation of notes for the two categories of rising tones and falling tones. With regard to the selection of offsets, we take the last frame of the former melody as the offset and add a new offset, starting with the second onset as the end of the previous note. Comparing the two methods, the only difference is in the threshold $\alpha$. Based on the temporal rhythm, our baseline $\alpha$ takes a value of 1. On the basis of our experiments, our extension $\alpha$ takes a value of 1.2.

### 3.3. Pitch Line Matching

The importance of pitch issues in note transcription has generally been ignored, and most of the criteria selected represent the mean value of pitch within the segmentation interval [35]. This method is certainly the most efficient; however, there are some deviations in the singing voice. RJ McNab [36] proposed a local pitch histogram to estimate the correct pitch. As shown in Figure 8, the existence of glide will affect the judgement of the entire pitch. Nevertheless, the stable area of singing voice mostly appears in the tail. Our proposed pitch line matching algorithm is as follows:

(1) Calculate the average pitch $\overline{P}$ of the note and round it to an integer MIDI pitch;
(2) Determine the five pitch lines $\overline{P}$, $\overline{P} \pm 1$, $\overline{P} \pm 2$, as shown by the dashed blue line in Figure 8;
(3) For the pitch $P_i$ of each frame, if the interval between a certain pitch line and $P_i$ is less than 50 cents, $P_i$ will be matched to that pitch line. Lastly, the most frequently matched pitches are recorded as the final pitch of the note;
(4) Repeat the above steps to determine the pitch of all notes.

We furthermore considered the characteristics of singing voices to avoid the interference of aspiration. This subtle improvement has a huge impact on our entire experiment process. Our proposed algorithm effectively improves the accuracy of pitch by over 4% on average.

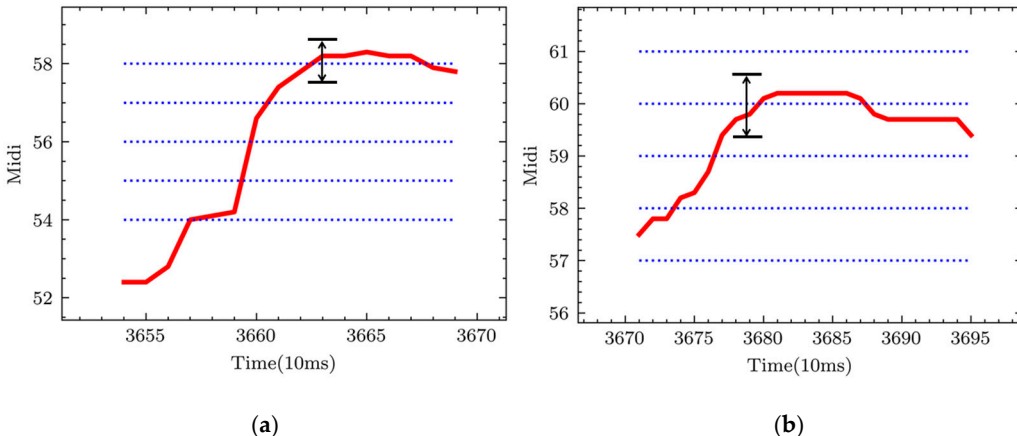

**Figure 8.** Using pitch line matching to obtain a pitch for each note. The red line is a note, the blue dashed lines are the pitch line, and the black arrow is the matching range (±50 cents); (**a**,**b**) are the fragment of the dataset "Cmedia".

## 4. Evaluation

### 4.1. Audio Melody Extraction

#### 4.1.1. Dataset

We used statistics from four classically used data collections from previous years that contain approximately 400 audio samples. The audio formats of all collections were wav files, the sampling rate was 44.1 KHz, the bit rate was 16, and only mono format was included. The datasets contained mostly vocal melodies, such as English songs, and matched all types of content, although some nonvoice audios was also included. Details of the test are as follows:

- MIREX09 database: 374 karaoke recordings of Chinese songs. Each recording was mixed at three different levels of signal-to-accompaniment ratio {−5 dB, 0 dB, +5 dB}, making a total of 1122 audio clips. Total time: 10,022 s.
- MIREX08 database: four excerpts of 1 min from "north Indian classical vocal performances". There are two different mixtures of each of the four excerpts with differing amounts of accompaniment, making a total of eight audio clips. Total time: 501 s.
- MIREX05 database: 25 phrase excerpts of 10–40 s from the following genres: rock, R&B, pop, jazz, solo classical piano. Total time: 686 s.
- ADC04 database: Dataset from the 2004 Audio Description Contest. A total of 20 excerpts of about 20 s each were used. Total time: 369 s.

#### 4.1.2. Metrics

There were four kinds of definition for the samples, which are explained as follows:

- TP: true positives. These were frames in which the voicing was correctly detected and where TPC means a correct pitch, TPCch means a chroma correct, and TPI means an incorrect pitch but truly voiced;

$$TP = TPC + TPCch + TPI \qquad (14)$$

- TN: true negative. These were frames in which the nonvoice was correctly detected;
- FP: false positive. These frames were actually unpitched but were detected as pitched;
- FN: false negative. These frames were actually pitched but were detected as unpitched.

All figures were evaluated in MIREX in terms of five metrics: overall accuracy (OA), raw pitch accuracy (RPA), raw chroma accuracy (RCA), voicing detection rate (VR), and voicing false alarm rate (VFA). The equations are defined below:

$$RPA, RCA = \frac{TPC, TPCch}{FN + TP} \qquad (15)$$

$$OA = \frac{TPC + TN}{TP + TN + FN + FP} \tag{16}$$

$$VD = \frac{TP}{TP + FN} \tag{17}$$

$$VFA = \frac{FP}{FP + TN} \tag{18}$$

For the correct voiced pitch ($f_0$) and chroma, we allowed tone buffers within $\pm \frac{1}{4}$ compared to the ground truth. For comparison, we selected the results from a competition from the past two years, which was dominated by results from 2019. The statistical result for each metric is the average of the algorithms over the presentation datasets.

### 4.1.3. Results

The algorithms from the results of MIREX for the past two years are shown below, and systems are denoted by the initials of their authors: KN4 [16], AH1 [37], HZ4 (our submission), HLD1 (different variants from AH1), KD1 [38], ZC1 [39], and BH (who submitted two variants) [40].

The overall accuracy of our algorithm (HZ4) was lower than that of KN4 and close to AH1 (Table 1). Both of the previous two algorithms were realized through neural network modeling. Our system focuses on the innovation of contour searching, in contrast to the traditional frame-to-frame filtering mode. Significantly, we directly analyzed the contours using existing datasets, especially the threshold values in each detection, which seemed to be acceptable. Due to the existence of a large number of thresholds in the experiment, there is still considerable room for progress in melody extraction. Owing to the limitations in the precision of MIREX, we only submitted the extension system.

**Table 1.** MIREX correlative results in 2019 and 2020.

| Algorithm | OA | RPA | RCA | VR | VFA |
|---|---|---|---|---|---|
| KN4 | **0.724** | 0.698 | 0.702 | 0.764 | 0.124 |
| AH1 | 0.703 | 0.746 | 0.772 | 0.797 | 0.171 |
| **HZ4** | 0.701 | 0.721 | 0.751 | 0.879 | 0.32 |
| HLD1 | 0.686 | 0.739 | **0.774** | 0.777 | 0.177 |
| KD1 | 0.681 | **0.749** | 0.767 | 0.832 | 0.373 |
| ZC1 | 0.671 | 0.612 | 0.619 | 0.698 | 0.174 |
| BH2 | 0.657 | 0.734 | **0.774** | 0.714 | 0.292 |
| BH1 | 0.64 | 0.721 | 0.756 | 0.707 | 0.312 |
| HH4 | 0.444 | 0.615 | 0.633 | 0.956 | 0.92 |
| WJH2 | 0.433 | 0.599 | 0.621 | 0.955 | 0.914 |

In addition to the estimation of accuracy, we also present the performance of each algorithm for different datasets, mainly including the statistical results for the overall accuracy (OA). As shown in Figure 9, in the ADC04 datasets, the algorithm proposed in this paper reached an optimal accuracy rate of 0.824; barring an accuracy rate of 0.815 for the KD1 algorithm, other algorithms did not reach a value of more than 80%. For the MIREX05 dataset, BH1 achieved the highest accuracy rate of 0.679, and the overall accuracy rate of this dataset was the lowest. For the MIREX08 and MIREX09 data sets, the highest accuracy rates were 0.793 and 0.828, respectively, and the results were within the applicable range.

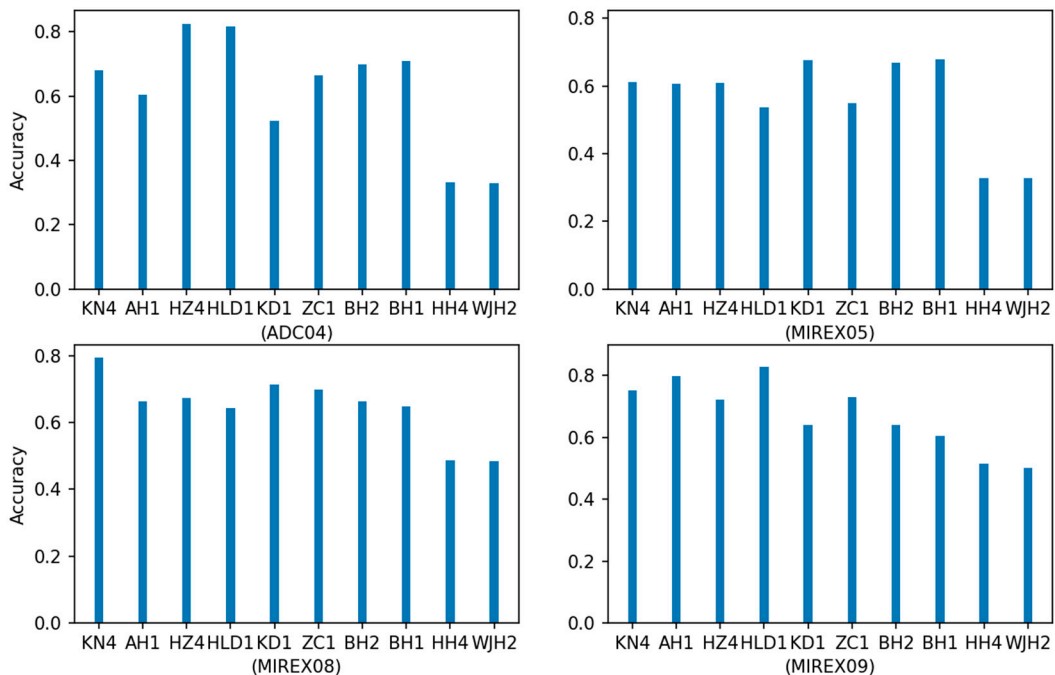

**Figure 9.** Overall accuracy (OA) results in each collection for an audio melody extraction task.

Figure 10 shows line graphs for the three parameters—VR, VFA, and OA—for different values of $\theta$ and $\mu$. Comparing (a) and (b), we can see from the curve fluctuation that the values of $\theta$ and $\mu$ are inversely proportional to the two parameters of VR and VFA; however, we paid more attention to the results of OA. When $\theta = 0.3$, OA reaches its highest value. It can be seen from the figure that the curve in Figure 10a is steeper, which proves that $\theta$ has a greater impact on the singing voice detection results. It can be seen from Figure 10b that, when $\mu$ is selected as 0.5, the OA is higher. This also shows the process of selecting parameters in our experiments.

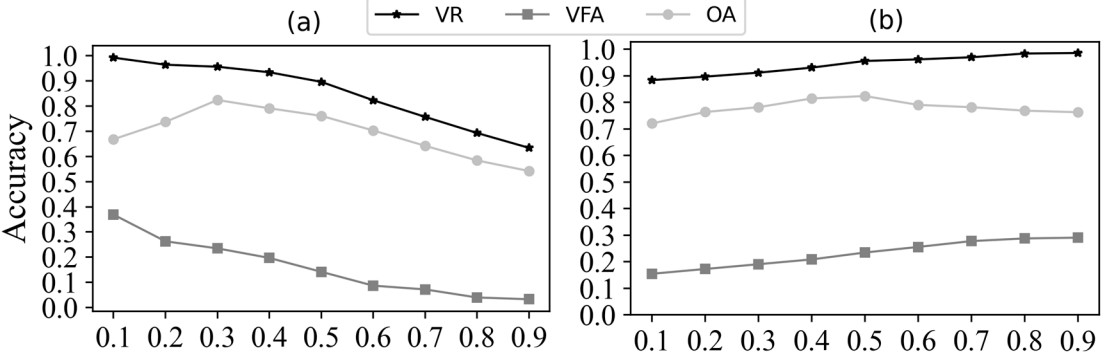

**Figure 10.** Different values of $\theta$ and $\mu$ for the singing voice results: (**a**) represents $\theta$ parameter influence and (**b**) represents $\mu$ parameter influence.

### 4.2. Singing Transcription

4.2.1. Dataset

A new mission presented online in MIREX2020 fitted the scope of our research exactly. The purpose of this task was to transcribe the singing voice from polyphonic music into a chain of notes, where each note was indicated by three parameters: the onset, offset and score pitch. Only one collection was used to evaluate the proposed system. The detailed description is as follows:

- Cmedia dataset: This dataset was composed of 200 YouTube links of pop songs (most are Chinese songs), including the ground truth files of the vocal transcription. In total, 100 audio files were released as the open set for training, and the other 100 were kept as the hidden set for testing.

This collection contained multiple accompanied melodies and one singing melody. The audio sampling rate was 44.1 KHz, the sampling size was 16 bits, and the dataset contained two-channel waves and mostly included Chinese songs. The training set can be downloaded at https://drive.google.com/file/d/15b298vSP9cPP8qARQwa2X_0dbzl6 _Eu7/edit (accessed on 21 May 2021).

### 4.2.2. Metrics

We evaluated the accuracy of the transcription by computing COnP and COn metrics [32], as well as computing the corrected transcribed notes of the ground truth. The following rules were utilized to determine whether notes were successfully matched:

- The onset difference was less than 100 ms in this competition;
- The pitch difference was less than 50 cents in this competition.

COnP requires the satisfaction of both conditions, while COn only requires the satisfaction of the first one. We computed the F-measure (FM), precision (Pr), and recall (Rec) on the overall results. The FM can be given by the following equation:

$$FM = 2 \cdot \frac{Pr \cdot Rec}{Pr + Rec} \tag{19}$$

### 4.2.3. Results

Our submission was HZ_SingingTranscription, which only contained the baseline system. We used mir_val [41] to evaluate our extension accuracy. Figure 11 shows the performance of our proposed two systems.

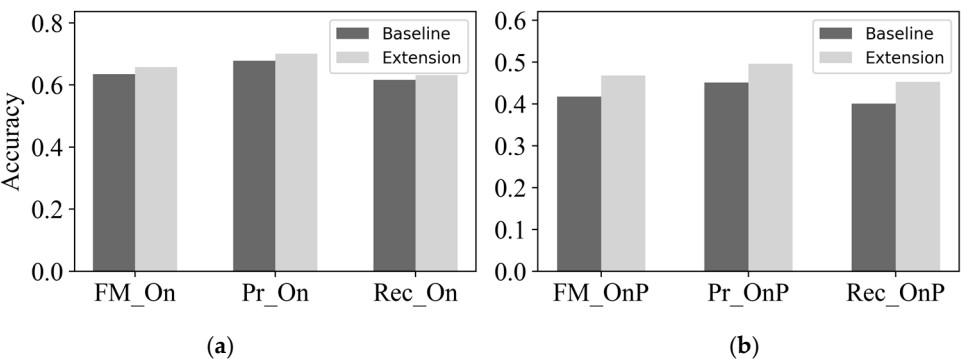

(a)                                    (b)

**Figure 11.** Singing transcription result from MIREX2020 in baseline system and extension system: (**a**) shows our on results, (**b**) shows our OnP results.

The result is obvious from the above description. In the pitch estimation, the advantage of the pitch line matching algorithm increased the overall accuracy of the extension to 0.468, which was about 5% higher than the accuracy of 0.411 of the baseline algorithm. Nevertheless, the consequence was that our programming time increased substantially; as a rough estimate, the time required doubled. There is apparently much room for improvement in distinguishing a singing voice from polyphonic music.

Therefore, it is more advantageous to calculate the pitch of a note for a more accurate extension pitch sequence. We readjusted the threshold of the onset difference to 150 ms. Figure 12 shows the accuracy results for the two algorithms within the new onset difference. It can be seen from the figure that the results of the two algorithms improved after adjusting the threshold, and the accuracy of the extension algorithm for OnP reached 0.556, which

was about 9% higher than the error range of 100 ms. The final result of the starting point detection also reached 0.702, breaking through the 70% accuracy level.

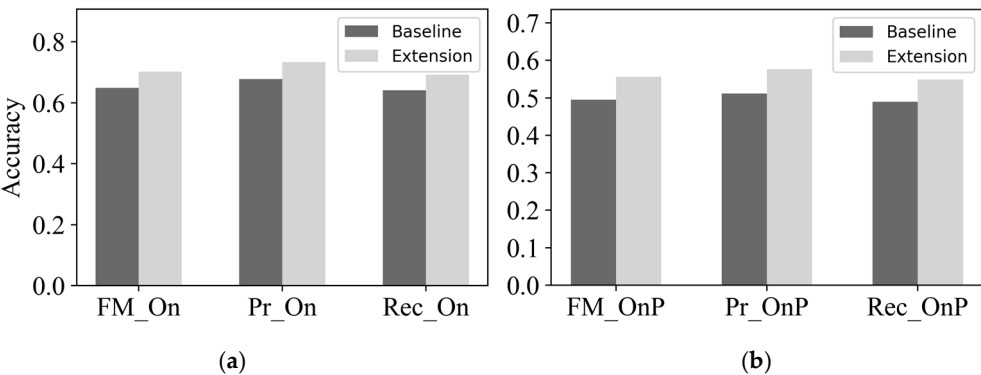

**Figure 12.** Overall result after 150 ms onset difference: (**a**) represents on results, (**b**) represents OnP results.

## 5. Conclusions

In this paper, we have proposed a method for singing transcription based on traditional signal processing and analysis, which involves the subclass method of melody extraction. In contrast to the current widely used neural network, our method aims to break down the content of each step and present the whole experiment in a clear and visible state. In addition, aspiration is mostly present at the beginning of a note being sung, leading to a greater backward deviation in onset detection. Non-voice sounds will have a lower energy component and lower frequency, and we thoroughly investigated the contours and analyzed their features by introducing the edge contour characteristics of the image, instead of being limited to the application of frame information. The main advantage of our method is the avoidance of multiple corrections of abnormal points, which lead to program redundancy, meaning we can enhance the efficiency of operation. The essence of using a Fourier transform to analyze contours lies in the characteristics of the contour; algorithms such as the sliding window, stabilization zone split note, and pitch line matching were introduced, and the importance of contour lines was suggested for the task of melody extraction from polyphonic music signals. For the selection of each parameter, our system used the results obtained from existing standard datasets, and multiple experiments were conducted to adjust the optimal parameters.

The accuracy of the algorithm in this paper depends greatly on the characteristics of the salience contour in the spectrogram. In many datasets, the accuracy and stability of the melody contour calculated by edge detection and contour filtering have been proven to be effective. However, in the singing voice detection task, there are still some shortcomings in terms of misjudgments and octaves. Among them, the low threshold value selected for human voice detection by this method preserves the outline of the human voice as much as possible while also retaining the outline of the accompaniment, which affects the accuracy of overall melody recognition. Therefore, it is hoped that a better signal processing method will be developed to distinguish between the different characteristics of the singing voice and its accompaniment.

**Author Contributions:** Conceptualization, Z.H.; methodology, Z.H.; software, Z.H.; validation, Y.F. and Z.H.; formal analysis, Y.F.; resources, Y.F.; writing—original draft preparation, Z.H.; writing—review and editing, Y.F.; project administration, Y.F. Both authors have read and agreed to the published version of the manuscript.

**Funding:** This research received no external funding.

**Institutional Review Board Statement:** Not applicable.

**Informed Consent Statement:** Not applicable. Written informed consent was obtained from patients to publish this paper.

**Data Availability Statement:** Data are available from https://www.music-ir.org/mirex/wiki/2020: Audio_Melody_Extraction (accessed on 21 May 2021) and https://www.music-ir.org/mirex/wiki/2020:Singing_Transcription_from_Polyphonic_Music (accessed on 21 May 2021).

**Conflicts of Interest:** The authors declare no conflict of interest.

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
