# Peer review of "Singing Transcription from Polyphonic Music Using Melody Contour Filtering"

_applsci, doi:10.3390/app11135913_

Round 1

Reviewer 1 Report

The paper presents a complex procedure for the automatic transcription of singing in polyphonic music, dividing it into two subtasks: melody extraction and note transcription.  Evaluation of the algorithm on MIREX datasets achieves high levels of transcription accuracy.

Only two points: first, a suggestion to better argument for the distinction into two subtasks.

Second, a thorough linguistic rephrasing because various English grammatical rules are disregarded, from absence of the verb "to be", to lack of plurals, to anacolutic sentences, and others.

Reviewer 2 Report

Please find the comments in the attachment.

Round 2

Reviewer 2 Report

I thank the authors for their work after the first round of reviews. The manuscript is so much improved in this new version.  I would like to raise only a few final minor questions:

  • Most of the bibliography links are broken. Please fix them.
  • Lines 222 and 223: “Excessive low-frequency energy is the main factor causing the potential deviation of the fundamental frequency.” Is this an observation pertaining your melody extraction method or is it consistently observed in the literature? In the first case, please, explain why does it happen. In the second, please, cite the source(s).
  • Lines 433 to 440: points 3a) and 3b) are conditions to be held, but what tasks are carried out if one of them holds? It seems like they have been accidentally omitted.
  • Formula 13: how come the incorrect pitches TPI are counted as true positives? By the way, expressions for the counts of the false positives, true negatives and false negatives are missing.
  • Table 2: why is the VFA so alarmingly high with your methodology?
  • Figure 10: parameters mu and theta are, as I understand it, a function of the spectra of each song, is this right? So this figures tell how accuracy indicators may vary from sample to sample according to each sample’s spectral features. Please explain this.

These will be my last comments before publication.

Kind regards.
